# Enhancing Therapeutic Efficacy and Safety of Immune Checkpoint Inhibition for Bladder Cancer: A Comparative Analysis of Injectable vs. Intravesical Administration

**DOI:** 10.3390/ijms25094945

**Published:** 2024-05-01

**Authors:** Pradeep Tyagi, Jason Hafron, Jonathan Kaufman, Michael Chancellor

**Affiliations:** 1Departments of Urology, University of Pittsburgh, Pittsburgh, PA 15260, USA; tyagip@upmc.edu; 2Michigan Institute of Urology, Troy, MI 48084, USA; 3Lipella Pharmaceuticals, Inc., Pittsburgh, PA 15208, USA

**Keywords:** bladder cancer, checkpoint inhibitor, PD1, MRI, liposome, drug delivery

## Abstract

Bladder cancer (BC) presents a significant global health burden, characterized by high recurrence rates post-initial treatment. Gender differences in BC prevalence and response to therapy emphasize the importance of personalized treatment strategies. While Bacillus Calmette–Guérin (BCG) remains a cornerstone of BC therapy, resistance poses a challenge, necessitating alternative strategies. Immune checkpoint inhibitors (ICIs) have shown promise, yet systemic toxicity raises concern. Intravesical administration of ICIs offers a potential solution, with recent studies demonstrating the feasibility and efficacy of intravesical pembrolizumab. Although systemic toxicity remains a concern, its localized administration may mitigate adverse events. Additionally, liposomal delivery of ICIs exhibits promises in enhancing drug penetration and reducing toxicity. Novel imaging modalities compatible with Vesical Imaging-Reporting and Data System (VI-RADS) and capable of predicting high-grade bladder cancer can aid the pre-operative shared decision making of patient and surgeon. Future research should focus on refining treatment approaches, optimizing dosing regimens, and leveraging advanced imaging techniques to improve patient outcomes. In conclusion, intravesical immunotherapy presents a promising avenue for BC treatment, offering enhanced therapeutic effectiveness while minimizing systemic toxicity. Continued research efforts are essential to validate these findings and optimize intravesical immunotherapy’s role in BC management, ultimately improving patient outcomes.

## 1. Introduction

Bladder cancer (BC) is the second most prevalent neoplasm of the urinary tract, with an estimated US prevalence of a million cases and an alarming recurrence rate of 75–80% after initial transurethral resection of exophytic tumor (TURBT) [1]. The average age at onset of BC is 74 years [2] with a fourfold higher prevalence of BC in males compared to females [3].

The interplay of innate and adaptive immune cells between genders potentially influences BC prevalence, prognosis, and response to immunotherapy, which underscores the necessity for tailored treatment approaches [3]. A lower BC prevalence in females [4] may be mechanistically linked to the antitumor effect triggered by recurrent stimulation of innate immune cells by bacteriuria [5] and the resultant urothelial exfoliation may be inadvertently getting rid of [6] precancerous urothelium cells in females. While a higher ratio of infiltrating innate and adaptive immune cells [3] lowers the overall BC prevalence in females, the same ratio of innate and adoptive immune cells affords an unfavorable response to immunotherapy for advanced BC culminating in lower progression-free survival (PFS) and overall survival of females with advanced BC [7].

Despite being the cornerstone of first-line therapy for BC, intravesical administration of Bacillus Calmette–Guérin (BCG) [8] is hindered by the resistance encountered in a significant proportion of patients [9,10]. BCG resistance is hallmarked by the expression of immunosuppressive molecules: programmed death (PD)L1, PDL2 ligands by BCG resistant tumor engages with the PD1 receptor on T cells to suppress anti-tumor response [10] (Figure 1).

In 2021, the US FDA approved injectable immune checkpoint inhibitors (ICIs) targeting the PD-1/PD-L1 interaction as a second-line treatment for cisplatin-ineligible patients with metastatic urothelial carcinoma [1]. These include the use of anti-PD1 checkpoint antibody (pembrolizumab) monotherapy for high-risk non-muscle invasive BC unresponsive to BCG [11] or together with enfortumab vedotin [12] as well as nivolumab [13], pembrolizumab in the platinum-refractory BC patients. However, the association of injectable ICIs with higher incidence of adverse events [14], see Table 1, motivates the question: whether the therapeutic index of IC1s could be improved via intravesical administration in BC patents [15].

## 2. Toxicity Associated with Intravenous Immune Checkpoint Inhibitors (ICIs)

The systemic breakdown of T-cell mediated immune surveillance by ICIs including pembrolizumab, resulting in immune-related adverse events such as pruritis with skin rash or scar covering <10% (Grade 1) to >30% (Grade 3–4) of body surface area, and endocrine toxicity of varying severity presents a significant challenge in the management of BC patients [16,18,21,22,23,24]. Guidelines from the American Society of Clinical Oncologist recommend monitoring of grade 1 toxicity and suspension of ICIs for most grade 2 toxicities with optional administration of steroids. Grade 3 treatment-related adverse events necessitate administration of steroids and/or antibodies targeting cytokines [14]. Grade 4 toxicity, occurring in 10% to 14% of nivolumab treated patients [13,23], requires permanent discontinuation of ICIs. Notably, treatment-related adverse events of grade 3 or higher occurred in 55.9% of enfortumab vedotin-pembrolizumab treated patients [25]. These adverse events stem from bystander effects of activated T-cells, an increase in levels of preexisting autoantibodies, elevated circulating inflammatory cytokines, and enhanced complement-mediated inflammation [1]. The potential for long-term adverse events with the rapid onset of action with intravenously injected IC1s underscores the importance of optimizing treatment strategies to mitigate systemic toxicity [1] and patient discomfort by slow onset of action with subcutaneous injection of ICIs [26].

In this regard, intravesical administration of ICIs offers a promising approach to increase the therapeutic index and reduce the systemic toxicity. A recent clinical study demonstrated the feasibility of intravesical pembrolizumab in combination with BCG induction therapy for BCG-unresponsive non-muscle invasive BC patients. Encouraging outcomes were observed, with notable improvements in recurrence-free survival and progression-free survival (PFS) [27]. The median age ranged from 76 to 82 years for patients receiving intravesical pembrolizumab (1–2 mg/kg) every 2 weeks from week 0 to week 17, and treatment period included a 5-week period of BCG induction with a weekly dose of intravesical BCG (TICE 50 mg) [17]. Weekly intravesical pembrolizumab (1–2 mg/kg) continued from week 18 to 49 at 4-week intervals. The elicited antitumor immune response was characterized by a significant increase in CD4+ T cells (as illustrated in Figure 1) and decreased expression of T-cell exhaustion markers in late recurrences. Although premature termination of the study due to pandemic-related reasons precluded the evaluation of the maximum tolerated dose, the study reported recurrence free rates at 6 and 12 months with the intravesical pembrolizumab dose (1–2 mg/kg) were 67% (95% confidence interval [CI]: 42–100%) and 22% (95% CI: 6.5–75%), respectively. While progression to locally advanced cancer occurred in 4 of 9 patients, PFS was 100% at 6 months, with a median PFS of 36 months.

## 3. Toxicity Associated with Intravesical Immune Checkpoint Inhibitors (ICIs)

Intravesical pembrolizumab [20], despite administration of two-fold higher number of doses than five weekly doses of BCG, demonstrated a lower incidence of grade 1–2 toxicity [17]. Most adverse events attributed to the five-week BCG treatment were bladder-related, with gross hematuria being the most common. Additionally, one patient receiving multiple intravesical doses of pembrolizumab (1–2 mg/kg) exhibited grade 5 toxicity of an autoimmune disorder (11.1%) and grade 2 dose-limiting toxicity of diarrhea lasting 21 days. While the number of patients in the study were few, grade 3–4 toxicity of intravesical pembrolizumab [17] was lower than that of intravenous pembrolizumab [25] in BC patients (Table 1). 

The antitumor immune response evoked by a single dose of pembrolizumab and its higher systemic toxicity in combination with BCG implicates that stark differences in the pharmacokinetics and pharmacodynamics of intravesical pembrolizumab and BCG were compounded by the differences in regimen of two entities. A higher systemic toxicity of intravesical pembrolizumab can be logically traced to a higher systemic absorption of a 200 times smaller molecule than BCG [28,29] as inverse size dependence of Stokesian diffusion principle dictates faster diffusion of smaller molecules and slower diffusion of macromolecules [30,31]. Moreover, the systemic uptake of pembrolizumab into bladder mucosa after intravesical administration for ≥30 min is assured by the tumoritropic infiltration of macromolecules of comparable size such as radiolabeled dextran-99mTechnetium conjugated to epidermal growth factor (EGF) after 30 min instillation in non-muscle invasive BC patients [32]. Not only is the Stokesian diffusion of macromolecules—BCG, pembrolizumab and EGF conjugated probe (Figure 1)—slower than that of smaller molecules like Mitomycin [molecular weight 334.3 Daltons, partition coefficient (log P)-0.4] [33,34,35] Gadobutrol [2,36] or fluorescein [37] but the extravasation of diffused macromolecules into capillaries and veins is slower than small molecules. As a result, fast venous clearance of Mitomycin [33,34], Gadobutrol [38] or Fluorescein [37] prevents their excessive buildup in the lesioned areas whereas slow lymphatic clearance of macromolecules [39] engenders a 2.4–1710-fold uptake of EGF conjugated probe relative to normal areas which reproduced the excessive buildup of labeled antibodies in transected lesions [32,40,41]. 

Accordingly, the clearance of diffused drugs from urothelium by venous outflow [42] keeps the urothelial levels of mitomycin [43] or of Gadobutrol static with time regardless of the duration of dwell time or repeated instillation [43,44,45]. The venous clearance of Mitomycin diffusing down the concentration gradient from bladder lumen [46] ensured that urothelial Mitomycin levels remain static during the instillation period [43] but the blood levels of instilled Mitomycin [34,47] and of Gadobutrol exhibited a dynamic range during and after instillation periods(Figure 2A). Since absorbed mitomycin is excreted renally, an increase in the systemic absorption of mitomycin [34] with longer dwell time of 1 h rather than 0.5 h in BC patients [34] can be ascertained from Mitomycin levels excreted in urine and higher urine levels of excreted Mitomycin coincided with a significant reduction of cancer recurrence [33,48]. The role of venous outflow in maintaining static urothelial drug levels after intravesical therapy is also corroborated by the blood levels of drugs in patients without BC [49,50] and by systemic absorption of radiolabeled urea, sodium and radio-iodinated albumin in accordance with the Stokesian diffusion principle [30,31] (see Figure 1). 

Therefore, the undetectable blood levels of pembrolizumab after intravesical administration [17] need to be reconciled with the quantification of 0.01% instilled dose of radio-iodinated albumin in the plasma of BC patients [30]. Based on pharmacokinetics of similar sized macromolecules, we estimate that the absorbed dose fraction of intravesical pembrolizumab is likely to be larger than 0.01% owing to BCG-evoked inflammation accelerating the paracellular diffusion but slower lymphatic clearance extending the pharmacokinetic and pharmacodynamic effects of pembrolizumab after single dose [32,39,40,41,47,51]. 

Given that maximum plasma levels of mitomycin at 40 min after instillation [47] in BC patients with larger tumors were nearly twice as high as BC patients with smaller tumors (plotted in Figure 2A), the reported variability in the systemic toxicity of intravesical pembrolizumab could be linked to the differences in the systemic uptake of pembrolizumab created by differences in size and grade of tumor [17]. Leaky tight junctions (Figure 2B) of undifferentiated cells in higher-grade tumor are amenable to a faster rate of paracellular diffusion of drugs which produced higher blood levels of mitomycin and thiotepa levels in BC patients with larger tumor [50]. Higher systemic uptake of drugs instilled in bladder of patients with larger bladder tumors is also corroborated by higher intravascular plus extravascular uptake of irrigation fluids with transurethral resection of larger sized prostate [52,53,54]. While I^131^ albumin (66.5 Kilo Daltons) admixed into irrigation fluids indexed the systemic intravascular absorption exclusively via injured blood vessels, two-fold higher uptake of I^125^ sodium iothalamate (0.635 Kilo Daltons; 100 times smaller than I^131^ albumin) reflected the paracellular ingress across leaky tight junctions plus the intravascular ingress of I^125^ sodium iothalamate [52,53,54]. The exclusive intravascular ingress of I^131^ albumin from resected prostate is hallmarked by much lower volume of distribution [55] than I^125^ sodium iothalamate.

The adverse impact of inflammation and transurethral bladder tumor resection (TURBT) on the systemic uptake of instilled drugs can also be gauged by the dramatic differences in the systemic uptake of instilled radio-iodinated iododeoxyuridine 24 h after TURBT relative to one to 4 weeks after TURBT [56]. The ratio for tumor to normal bladder wall uptake of radio-iodinated iododeoxyuridine was reported to be higher than other small molecules. Usually, chemotherapeutic drugs are instilled after TURBT, but a recent study inserted an intravesical drug delivery device for the sustained delivery of Gemcitabine (molecular weight 263.19 Daltons, log P-1.4) both before and after TURBT in non-muscle invasive BC patients [57]. While sustained delivery produced peak Gemcitabine urine levels by the third day of administration, insertion and removal of the device is required. Efficacy and cost benefit over instillation of mitomycin for 60 min [33,48] remains to be shown. Sustained delivery of lidocaine demonstrated limited efficacy for patients with interstitial cystitis/bladder pain syndrome [58]. While rapid venous clearance of small molecules like lidocaine and Gemcitabine from urothelium could be insurmountable with sustained delivery, macromolecules instilled in bladder can leverage the slow lymphatic [39] clearance to mount significant changes in the tumor microenvironment two weeks after a single intravesical dose of pembrolizumab in BC patients [17]. The delayed lymphatic clearance of another macromolecule, onabotulinum toxin may have extended the duration of decreased urgency episodes in association and lowered urothelial P2X3 expression for one month after just single instillation [59]. The echo of sustained delivery can also be heard in the intravesical delivery of Nadofaragene firadenovec which transfects urothelial cells by non-replicating adenovirus vectors for a sustained expression of human interferon alfa-2b gene to target tumor growth [60]. 

While the systemic absorption of intravesical pembrolizumab could be heightened by immature tight junctions of undifferentiated high-grade tumors and carcinoma in situ but the low assay sensitivity and inefficient timing of blood sampling could still contribute to the undetectable blood levels of pembrolizumab in BC patients exhibiting characteristic ICI toxicity after intravesical pembrolizumab [17]. In summary, accumulating clinical evidence and quantitative analysis of excised tumors from BC patients support the potential of intravesical ICIs as a promising approach to improve PFS while reducing recurrence and adverse events in BC patients.

## 4. Novel Imaging Modalities for Early Diagnosis of Muscle Invasive Bladder Cancer

Given the pivotal role of structural determinants of UC in erecting the blood-urine barrier (Figure 3) and of the tumor microenvironment in tumor progression, the uptake of instilled dyes (Gadobutrol) can serve as a virtual surrogate for monitoring tumor progression and treatment response. In pursuit of this objective, we have recently developed a radiation-free approach to monitor BC progression and invasion using intravesical contrast-enhanced magnetic resonance imaging (ICE-MRI) [2]. ICE-MRI may be also able to quantify the differences in bladder inflammation before and after intravesical BCG and pembrolizumab by documenting the reduced systemic uptake of instilled Gadobutrol [38]. 

To evaluate the ability of ICE-MRI to quantify treatment-associated reduction of tumor volume, we conducted experiments in mice fed 0.05% N-butyl-N-(4-hydroxybutyl) nitrosamine (BBN) water for 12 weeks [33,34] (see Figure 3). Tumor volume was assessed before treatment at week 11 and seven weeks after single intravesical dose treatment of either vehicle (water) or pembrolizumab at 0.3 mg/kg [61]. The substantial reduction in tumor volume observed in the pembrolizumab-treated group compared to mice receiving vehicle is substantiated by significant changes in the tumor microenvironment noted two weeks after a single intravesical dose of pembrolizumab in BC patients [17]. The findings of intravesical pembrolizumab-treated BBN tumor in mice (Figure 3) could be bridged to a median decrease of 41% in urine levels of granulocytes two weeks after a single intravesical dose of pembrolizumab in association with a significant median increase of 18% and 12% in CD4+ and CD8+ T cells, respectively of BC patients [17].

Given the significant treatment effect of a single intravesical dose of pembrolizumab in mice (Figure 2), it is reasonable to evaluate the clinical efficacy of pembrolizumab in absence of BCG [17]. Moreover, the absence of BCG-evoked inflammation may allow for safe administration of even higher doses of intravesical pembrolizumab at 5 mg/kg at a longer interval of 4 weeks as the half-life of pembrolizumab is only 26 days. The accrued clinical experience with intravesical BCG in the management of UC, alongside preliminary preclinical data (Figure 3) and pilot clinical studies on intravesical pembrolizumab [17], provides a roadmap for mitigating the long-term and potentially irreversible adverse events associated with injectable ICIs (Table 1). Future research directions include optimizing dosing regimens, exploring novel drug delivery approaches, and harnessing advanced imaging techniques to monitor treatment response and minimize systemic toxicity [2,36]. 

## 5. Liposomal Delivery of PD-1/PD-L1 and Biologics for Bladder Cancer

The concerns over the therapeutic index of injectable ICIs and the treatment success of intravesical BCG underscores the importance of intravesical administration for safe induction of localized anti-tumor response by ICIs in alignment with the principle of “bladder before blood” (BBB) [62], particularly for drugs with lower therapeutic index [49]. The uptake of dyes and drugs along umbrella cell borders may be more sensitive to the inflammation than malignancy as chemokines released by inflammatory foci of interstitial cystitis patients dilated tight junctions to dramatically raise the systemic uptake of instilled radiolabeled probes [30] and thiotepa [50] relative to BC patients. Therefore, avoiding BCG is likely to reduce the contribution of inflammation [38] in the systemic uptake and the resultant systemic toxicity [63] of intravesical pembrolizumab. 

Apart from avoiding BCG, another option to reduce systemic toxicity profile of intravesical pembrolizumab could be the liposome entrapment as reported for liposomal doxorubicin [64,65] and other macromolecules such as oligonucleotides [63], small interfering RNA, and onabotulinum toxin [59]. Liposome entrapment is expected to reduce the systemic toxicity via reduction in systemic uptake and an increase in the tumor localization of pembrolizumab. In addition, the observed effects of liposomes on urinary symptoms suggest that the biophysical properties of liposomes composed of endogenous lipids [63], can effectively mitigate the localized toxicity associated with ICIs. 

Although in vitro studies on cultured bladder epithelial cells [66] have demonstrated endocytosis of liposomes across the apical surface, detailed examination of rodent bladder harvested eight hours after instillation of fluorescent liposomes in vivo were [67] not consistent with the transcellular entry of liposomes. Instead, rat bladder post-instillation of fluorescent liposomes reproduced the dark apical surface of UCs along with fluorescent UC borders seen with confocal laser endomicroscopy of human bladder after brief instillation of Fluorescein [37] (see Figure 2B). Fluorescent borders of UC suggests that empty liposomes and liposomes complexed with small molecules or macromolecules [59] are more likely to enter the bladder via UC borders than through the apical surface of UC lined with asymmetric unit membrane (AUM) (Figure 1 and Figure 3). This discrepancy underscores the importance of translational research in drug delivery and cautions against premature conclusions drawn solely from in vitro studies on cultured cells. The limitations of cultured cells in replicating the transcellular barrier of AUM [68] hinders a comprehensive mechanistic understanding of infection and the in vitro and in vivo correlation of intravesical chemotherapy [69], as demonstrated by the efficacy of paclitaxel in BC [19,62]. Furthermore, static exposure of Gadolinium chelates like Gadobutrol especially with undifferentiated cells in-vitro [70] generated erroneous evidence for intracellular entry of 10–20 ng per thousand which is contrary to the poor intracellular entry of Gadolinium chelates with dynamic exposure in vivo [71], hallmarked by 3-fold lower volume of distribution (Vd) Liters (L) per kg of body weight relative to Vd of water. While Vd of 0.6 L/kg for water represents facilitated entry of water into all body compartments including intracellular, extracellular, and intravascular space, Gadobutrol with Vd of 0.23 L/kg represents extravasation of non-protein bound chelate into extracellular space and not into the intracellular space [72]. 

Therefore, static drug levels in-vitro cannot predict the in vivo pharmacology of diffused molecules after intravesical administration as mucosal blood flow ensures that plasma concentration is dynamically variable with time. Since urine levels are a lagging indicator of plasma levels, urine levels are an unreliable indicator of systemic absorption when the dwell time is longer than 5 min for instilled drugs [33,48]. Considering the success of intravenous chemotherapy when combined with intravenous immunotherapy for platinum-refractory BC [19,62], it is conceivable that BC patients with a poor prognostic gene signature [73] may derive additional benefit from a combination of intravesical pembrolizumab and chemotherapeutic agents [62].

## 6. Conclusions

In conclusion, the comparative analysis between injectable [1,25,26] and intravesical [17] administration of ICIs underscores the theoretical advantages in enhancing the therapeutic effectiveness while minimizing systemic toxicity in treating BC. Intravesical delivery of PD-1/PD-L1 inhibitors showed promise in mitigating long-term adverse events associated with injectable ICIs. Moreover, the intravesical liposomal delivery platform presents a potentially safe and practical method for delivering therapeutics to the urinary bladder. Additionally, we introduced a novel imaging modality, ICE-MRI, for early diagnosis of muscle-invasive bladder cancer based on tumoritropic infiltration of paramagnetic dyes [2] instead of colored dyes [74] with or without the conjugation to agents capable of targeting the metabolic programming of immune [75] and cancer cells [76]. Novel imaging modalities compatible with Vesical Imaging-Reporting and Data System (VI-RADS) and capable of predicting high-grade bladder cancer can facilitate the pre-operative shared decision making of patient and surgeon [77] and help predict the risk of BCG failure [78]. Further research and clinical trials are imperative to validate these findings and refine the treatment approaches for improved patient outcomes. These efforts are vital for advancing our understanding of intravesical immunotherapy and optimizing its role in the management of bladder cancer.

## Figures and Tables

**Figure 1 ijms-25-04945-f001:**
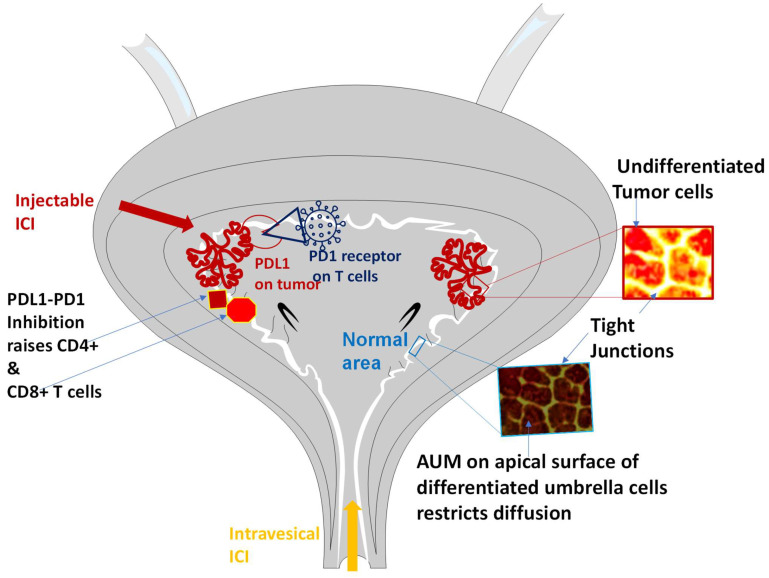
Schematic illustration of injectable ICI and intravesical ICI in blocking the innate immune evasion of bladder tumors stemming from the interaction of programmed death (PD) ligand PD-L1 released by tumor with PD-1 receptor expressed by T immune cells. Regardless of the route of administration, pharmacological blockade of PDL1/PD-1 interaction awakens the innate anti-tumor response hallmarked by a significant rise in CD4+ and CD8+ cells in bladder tumor. Intravesical ICI seeks to achieve the same treatment goal without incurring the treatment emergent life-threatening toxicity of injectable ICI by using the same route used for awakening antitumor response with BCG instillation. Umbrella cells (UC) and structural determinants of blood urine barrier erect a surmountable barrier to entry for intravesical ICI. The apical surface of differentiated UC lined with asymmetric unit membrane (AUM) erects a water-tight lining that restricts the transcellular entry of urine constituents as well as instilled drugs and macromolecules including BCG and antibodies, but immature tight junctions of undifferentiated tumor cells accelerate the size dependent passive, paracellular, Stokesian diffusion of drugs along UC borders. While faster diffusion of small molecules gets accelerated by the faster venous clearance of small molecules (a faster push and pull in the same direction), size dependent slower diffusion of macromolecules gets compounded by the slower lymphatic clearance of macromolecules (slower push and pull in same direction) to generate 5–50-fold higher tumoritropic infiltration of macromolecules (BCG) than small molecules (mitomycin) relative to normal areas in specimens excised by TURBT or cystectomy.

**Figure 2 ijms-25-04945-f002:**
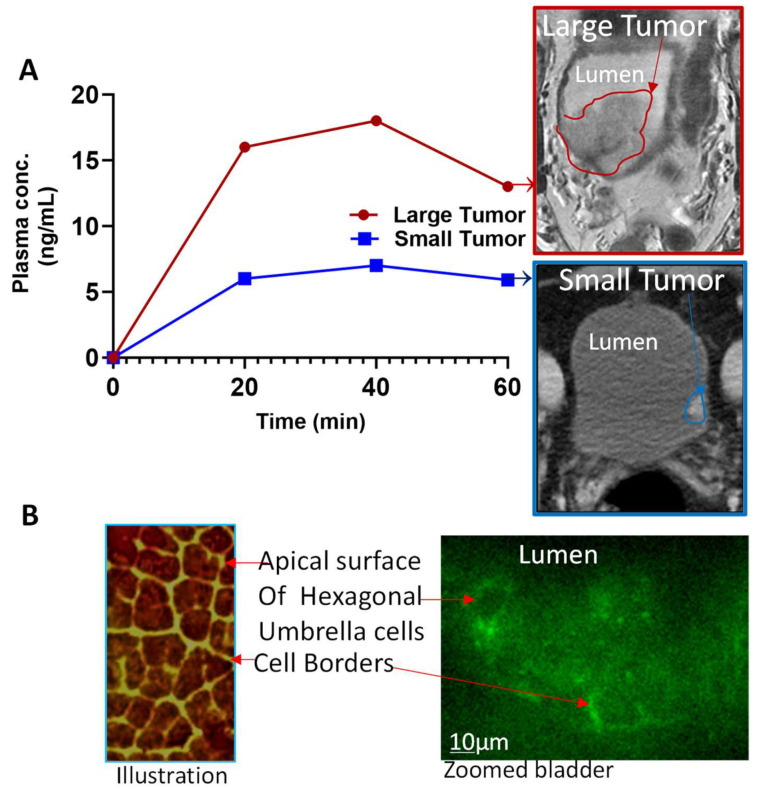
Panel (**A**): Enlarged tumor elevates the blood levels of instilled chemotherapeutic drugs (small molecules) owing to augmented paracellular diffusion across leaky tight junctions of undifferentiated cells and hypoxia ensuing from nutrient scarcity. Plasma drug levels in patients for tumors imaged by T1 weighted DCE-MRI are projected based on published clinical studies on mitomycin. Panel (**B**): Paracellular entry of chemotherapeutic drugs can be visualized by the fluorescent umbrella cell borders of rat bladder seen after instillation of fluorescent liposomes, recapitulating the luminal surface view of human bladder on cysto confocal endomicroscopy. The dark apical surface of polygonal umbrella cells represents the restricted transcellular diffusion whereas the size dependent paracellular traffic is displayed by the bright umbrella cell borders.

**Figure 3 ijms-25-04945-f003:**
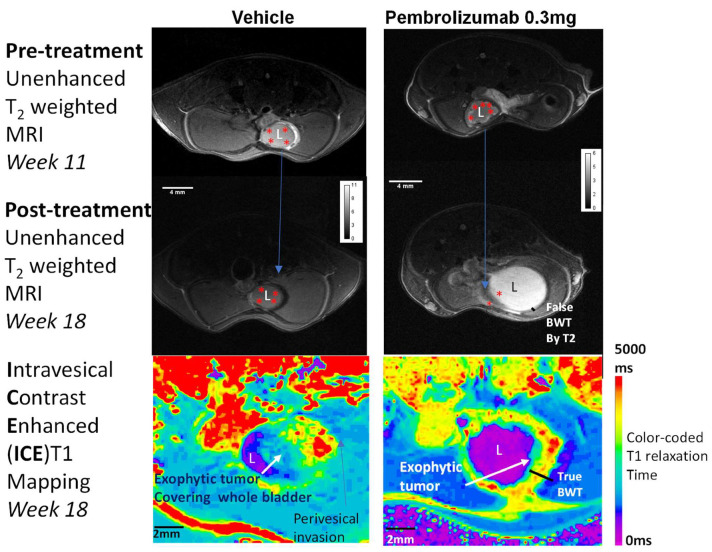
Virtual monitoring of antitumor response evoked by single intravesical dose of Pembrolizumab 0.3 mg administered by 24 G transurethral catheter to female mice fed 0.05% BBN water for 12 weeks. Tumor (red *) size was virtually measured by T2 weighted MRI just prior to treatment on week 11 and by multiparametric MRI on week 18 (post-treatment). Since bright signal of urine within lumen (L) obscures the true urothelial and bladder wall thickness (BWT) in T2 weighted MRI, the effect of single intravesical dose of Pembrolizumab on true BWT is displayed by the color-coded voxel-wise mapping as paracellular diffusion of instilled Gadobutrol into urothelium shortens T1-relaxtaion time from ~3500 ms measured prior to instillation to ~1000 ms after instillation and just before animal sacrifice on week 18. While tumor (red *) growth was comparable for two groups on week 11, Pembrolizumab treatment dramatically reduced the tumor volume compared to vehicle treatment, as displayed by the size of purple tumor-free bladder lumen in bottom row of two groups.

**Table 1 ijms-25-04945-t001:** Incidence of treatment related adverse events with ICIs.

Adverse Event	Injectable Pembrolizumab (*n* > 500)		Intravesical Pembrolizumab (*n* = 9)	
Pruritis	3.2–10%	[16]	11%	[17]
Endocrine toxicity	6.5–37%	[18,19]	-	[20]
Pneumonitis	4.9%	[18]	-	
Grade 1–2 urinary symptoms	11%	[8]	44%	[17]
Grade 2 diarrhea	11–20%	[19]	22.2%	[17]
Fatigue, arthralgia	33.3%	[19,21]	22.2%	[17]
Immune-related adverse events	66%	[19,22]	11.1%	[17]

Smaller sample size of BC patients (*n* = 9) treated with intravesical ICI in just one study precludes a reasonable comparison of toxicity observed in BC patients treated with intravenous ICIs in several cited studies which are indicated by the reference number inside square brackets.

## Data Availability

Data can be made available upon request to authors.

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
