# Peer review of "Enhancing Therapeutic Efficacy and Safety of Immune Checkpoint Inhibition for Bladder Cancer: A Comparative Analysis of Injectable vs. Intravesical Administration"

_ijms, 2024, doi:10.3390/ijms25094945_

Round 1
Reviewer 1 Report
Comments and Suggestions for Authors
This review adresses an interesting topic of ICI administration. Several improvements are required.
1. Following sentence is not relevant in the absract:
Novel imaging modalities, such as Intravesical Contrast-Enhanced Magnetic Resonance Imaging (ICE-MRI), offer early diagnosis of muscle-invasive bladder cancer, aiding in treatment monitoring.
2. "These include the use of anti-PD1 checkpoint antibody 60
(pembrolizumab) alone or together with enfortumab vedotin [10] as well as nivolumab 61
[11], pembrolizumab, or avelumab [12] in the platinum-refractory BC patients." This is not quite true - avelumab is suitable for pts with response or SD following platinium.
3. This is out of context "However, 62
the use of injectable ICIs may be associated with a higher incidence of adverse events 63
compared to BCG [13], raising concerns regarding the optimal method and formulation 64
of intravesical administration and their therapeutic index [14]." In previous sentence you described advanced BC in which BCG has no role. Please correct because these sentences are misleading.
4. Please cite and mention the name of trials confirming this sentence :Recent clinical studies have 82
demonstrated the feasibility and efficacy of intravesical pembrolizumab in combination 83
with BCG induction therapy for BCG-unresponsive non-muscle invasive BC patients. Efficacy results are not yet avaialable and this should be mentioned
5. Please mention the most common toxixicites of ICI administered systematically. I cannot see that in the text.
6. Please explain why tumor size matters in terms of treatment toxicity? Given that maximum plasma levels of mitomycin 130
recorded at 40min after instillation [30] nearly doubled in BC patients with larger tumors 131
compared to BC patients with smaller tumors in a clinical study, the variability in the 132
systemic toxicity of intravesical pembrolizumab could stem from the differences in rate 133
and extent of systemic uptake due to differences in the tumor size and grade of enrolled 134
patients [18].
I believe that pembro should not be given when tumor is not removed (before TURBT).
7. Please explain the idea of using Gadobutrol and how this is relevant to ICI?
Higher systemic absorption of mitomycin [25] with longer dwell 148
time is affirmed by a significant reduction of recurrence [24] coinciding with higher urine 149
levels of mitomycin. Importantly, mucosal blood flow keeps urothelial drug levels or 150
Gadobutrol static with time regardless of the duration of dwell time or with repeat instil- 151
lation [33-35] whereas blood levels of instilled mitomycin [25,30] and Gadobutrol vary 152
with time.
8.Suggestion of BCG avoidance has no scientific grounds. "Apart from avoiding BCG, another option to reduce systemic toxicity" Please comment on that. Are there any trials BCG vs pembro underway?
9. What about subcutenous administration of ICI and its toxicity? such discussion is lacking
10. "In conclusion, the comparative analysis between injectable [1,16] and intravesical [18] 248
administration of ICIs underscores the potential superiority of the la􀄴er in enhancing ther-" I cannot agree with superiority of intravesical ICI. Please be specific, name trials and evidence and disease stage.
11. A table with comparison of most imporant/frequent adverse events between injectable and intravesical ICI is required. This is the main topic of the paper and is not accurately adressed.
12. Consider mentioning other papers in the topic:
DOI:https://doi.org/10.1016/S1470-2045(21)00147-9 - trial showing pembro as alternative to cystectomy in BCG unresponsive NMIBC
- DOI: 10.1016/S1470-2045(20)30540-4 - trial showing alternative to cystectomy in BCG unresponsive NMIBC
- DOI: 10.1007/s11255-019-02183-5 - review mentioning many ICI trials in NMIBC in the context of BCG treatment
Author Response
Dear Editor:
We would first want to thank the reviewers for the positive review and for their insightful comments that make this a stronger manuscript. We have made all the recommended changes and addressed below. We hope the manuscript is now acceptable for publication. Thank you.
Reviewer 1
This review addresses an interesting topic of ICI administration. Several improvements are required.
- Following sentence is not relevant in the abstract: Novel imaging modalities, such as Intravesical Contrast-Enhanced Magnetic Resonance Imaging (ICE-MRI), offer early diagnosis of muscle-invasive bladder cancer, aiding in treatment monitoring.
Response: We thank the reviewer for critical review of our manuscript, and we agree with the relevance criteria for imaging. Accordingly, the sentence highlighted by reviewer is now replaced by: Novel imaging modalities compatible with Vesical Imaging-Reporting and Data System (VI-RADS) and capable of predicting high-grade bladder cancer can aid the preoperative shared decision making of patient and surgeon.
- "These include the use of anti-PD1 checkpoint antibody (pembrolizumab) alone or together with enfortumab vedotin [10] as well as nivolumab[11], pembrolizumab, or avelumab [12] in the platinum-refractory BC patients." This is not quite true - avelumab is suitable for pts with response or SD following platinum.
Response: We thank the reviewer for the comment, and we have now removed avelumab from the sentence.
- This is out of context "However, the use of injectable ICIs may be associated with a higher incidence of adverse events compared to BCG [13], raising concerns regarding the optimal method and formulation of intravesical administration and their therapeutic index [14]." In the previous sentence you described advanced BC in which BCG has no role. Please correct because these sentences are misleading.
Response: We agree that using BCG as reference for toxicity for ICIs was not advisable. We have now revised the sentence to improve clarity.
- Please cite and mention the name of trials confirming this sentence :Recent clinical studies have demonstrated the feasibility and efficacy of intravesical pembrolizumab in combination with BCG induction therapy for BCG-unresponsive non-muscle invasive BC patients. Efficacy results are not yet available, and this should be mentioned
Response: We thank the reviewer for this comment, and we have removed efficacy after “ feasibility” in the sentence. Although the pandemic caused premature termination of trial, data accrued till termination generate proof of principle and efficacy analysis revealed a modest response, with 22% of patients remaining recurrence-free at 1 yr.
- Please mention the most common toxicities of ICI administered systematically. I cannot see that in the text.
Response: Thank you and we have now mentioned common toxicities (pruritis, colitis, hepatitis, pneumonitis, myositis and endocrine toxicity) and cited multiple sources for Grade 1 to 4 toxicity information on ICIs and the guidance to manage such toxicity in the paper.
- Please explain why tumor size matters in terms of treatment toxicity? Given that maximum plasma levels of mitomycin recorded at 40min after instillation [30] nearly doubled in BC patients with larger tumors compared to BC patients with smaller tumors in a clinical study, the variability in the systemic toxicity of intravesical pembrolizumab could stem from the differences in rate and extent of systemic uptake due to differences in the tumor size and grade of enrolled patients [18]. I believe that pembro should not be given when tumor is not removed (before TURBT).
Response: We thank the reviewer for asking highly pertinent questions and we have revised the text on this topic to improve clarity. Multiple studies of last 50 years have validated that systemic toxicity of intravesical chemotherapy is determined by blood levels of instilled drugs. Comparative studies between BC patients with small and large tumor established significantly elevated peak blood levels of mitomycin and thiotepa in patients with large tumor because leaky tight junctions of undifferentiated cells in larger and higher-grade tumor permit a faster rate of paracellular diffusion of drugs. We agree with the reasonable suggestion of giving Pembrolizumab after TURBT but intravesical treatment prior to TURBT (PMC10998271) can reduce the size of tumor for resection as some harbor the fear that TURBT of larger size risks seeding cells for recurrence.
- Please explain the idea of using Gadobutrol and how this is relevant to ICI? Higher systemic absorption of mitomycin [25] with longer dwell time is affirmed by a significant reduction of recurrence [24] coinciding with higher urine levels of mitomycin. Importantly, mucosal blood flow keeps urothelial drug levels or Gadobutrol static with time regardless of the duration of dwell time or with repeat instillation [33-35] whereas blood levels of instilled mitomycin [25,30] and Gadobutrol vary with time.
Response: As stated in response to previous comment, we have now substantially revised the text for clarity and to explain the relevance of venous and lymphatic system of bladder in clearing diffused drugs. Since MRI can virtually track the transit of instilled Gadobutrol in bladder wall from lumen to urothelium and veins, the transit of instilled Gadobutrol can shed light on the transit of mitomycin or Pembrolizumab (ICI) instilled in bladder. Line 179
8.Suggestion of BCG avoidance has no scientific grounds. "Apart from avoiding BCG, another option to reduce systemic toxicity" Please comment on that. Are there any trials BCG vs pembro underway?
Response: We thank the reviewer for providing us the opportunity to provide context for our suggestion of BCG avoidance. For BCa patients reporting recurrence post-BCG or unresponsive to BCG, we were suggesting that avoiding the BCG evoked inflammation in bladder may reduce the systemic absorption of intravesical Pembrolizumab and thereby its systemic toxicity. While we are not aware of any trials of intravesical Pembrolizumab, Keynote 676 trial is underway on intravesical BCG in combination with systemic pembrolizumab for recurrent or persistent high-grade non-muscle invasive bladder cancer.
- What about subcutaneous administration of ICI and its toxicity? such discussion is lacking
Response: We thank the reviewer for the insightful comment. Yes, the landscape of ICI is constantly evolving with recent approval of atezolizumab for subcutaneous administration which can cut down the injection time, decrease patient discomfort, and the possibility of self-administration akin to insulin or EpiPen injection.
- "In conclusion, the comparative analysis between injectable [1,16] and intravesical [18] administration of ICIs underscores the potential superiority of the la?er in enhancing ther-" I cannot agree with superiority of intravesical ICI. Please be specific, name trials and evidence and disease stage.
Response: We agree that use of the term superior was premature at this stage, and we have replaced it with theoretical advantages.
- A table with comparison of most important/frequent adverse events between injectable and intravesical ICI is required. This is the main topic of the paper and is not accurately addressed.
Response: We thank the reviewer for constructive suggestion, and we have now added a table
- Consider mentioning other papers in the topic:
DOI:https://doi.org/10.1016/S1470-2045(21)00147-9 - trial showing pembro as alternative to cystectomy in BCG unresponsive NMIBC
- DOI: 10.1016/S1470-2045(20)30540-4 - trial showing alternative to cystectomy in BCG unresponsive NMIBC
- DOI: 10.1007/s11255-019-02183-5 - review mentioning many ICI trials in NMIBC in the context of BCG treatment
Response: We thank the reviewer for the suggested references, which are now included as references 11, 51 and 65 in the revised manuscript.
Reviewer 2 Report
Comments and Suggestions for Authors
The communicaton is interesting but several errors need to be corrected before it may be accepted for publication:
1. Figure 1 is misleading - it shows two different styles of graphics in a single image - one futuristic presumably produced with AI and the other one "simple paint schemes" which should be corrected.
2. English seems not fitting the nature of publication style of writing. It seems very upscale and does not sound scientific.
3. Figure 2 is a table and should be revised.
4. In fig. 3 the left part is describes as illustration which is quite a general statement and in fact all the figures are illustrations.
Comments on the Quality of English Language2. English seems not fitting the nature of publication style of writing. It seems very upscale and does not sound scientific.
Author Response
Dear Editor:
We would first want to thank the reviewers for the positive review and for their insightful comments that make this a stronger manuscript. We have made all the recommended changes and addressed below. We hope the manuscript is now acceptable for publication. Thank you.
Reviewer 2
The communication is interesting, but several errors need to be corrected before it may be accepted for publication:
- Figure 1 is misleading - it shows two different styles of graphics in a single image - one futuristic presumably produced with AI and the other one "simple paint schemes" which should be corrected.
Response: We thank the reviewer for insightful comment and yes there are indeed two different styles of graphics . Guided by the axiom that a picture can counter a thousand words of skepticism, we used illustration for conveying the main point of intravesical drug delivery that runs against the pervasive dogma of water-tight bladder lining. We have now revised the figure and legend to illustrate the significance of paracellular diffusion between cell borders and the advantage of delayed lymphatic clearance extend the biological effects of macromolecules like BCG or antibody without the need of sophisticated sustained release systems needed for small molecules like Gemcitabine which are cleared by venous blood from urothelium.
- English seems not fitting the nature of publication style of writing. It seems very upscale and does not sound scientific.
Response: We thank the reviewer for the comment and would like to affirm that native speakers of English have checked the text for clarity.
- Figure 2 is a table and should be revised.
Response: We would like to clarify the rearrangement of figures in revised manuscript. The new Figure 3 corresponds with Figure 2 of original manuscript, which is a representative data on mouse bladder imaging by intravesical contrast enhanced (ICE)-MRI. Figure 2 displays axial images of mouse abdomen and T1 mapping of the bladder wall to substantiate paracellular diffusion of instilled Gadobutrol for intravesical contrast enhanced MRI. We have now added a Table for comparing toxicity of intravenous and intravesical ICIs.
- In fig. 3 the left part is describing as an illustration which is quite a general statement and in fact all the figures are illustrations.
Response: Per our previous comment, Figure 2 of original manuscript is Figure 3 of revised manuscript. Figure 3 contains graphs and DCE-MRI images of human subjects with small and large tumors as well images of rat bladder cryosections taken after instillation of fluorescent liposomes. To communicate the technical message of intravesical delivery of antibodies to a broader audience, we relied on illustrations to improve comprehension. Importantly, illustrations visually match the pictures taken by confocal endomicroscopy of human bladder available in reference 31.
Round 2
Reviewer 1 Report
Comments and Suggestions for Authors
I cannot agree that "These include the use of anti-PD1 checkpoint antibody 67 (pembrolizumab) monotherapy for muscle invasive BC unresponsive to BCG[11]"
monotherapy is not suitable for MIBC, this is a mistake, and I guess that you ment NMIBC
please correct
Author Response
Dear Editor:
We thank the reviewers for their constructive comments on our manuscript. We have made all the requested changes. The manuscript has been now revised and changes in text are highlighted in yellow and the response to specific comments is given below:
We hope the manuscript is now acceptable for publication. Thank you.
Reviewer 1
I cannot agree that "These include the use of anti-PD1 checkpoint antibody 67 (pembrolizumab) monotherapy for muscle invasive BC unresponsive to BCG[11]"monotherapy is not suitable for MIBC, this is a mistake, and I guess that you mean NMIBC please correct
Response: We thank the reviewer for identifying that error of missing “ non” before muscle and we have now rectified that in revised manuscript.
Reviewer 2:
The manuscript was not improved enough.
- Figure 1 still doesn’t look professional.
Response: Thank you and we have replaced Figure 1 to convey the main message of intravesical immunotherapy of bladder cancer by checkpoint inhibitors.
- Table 1 - it is hard to understand the Xs.
Response: We thank the reviewer for a constructive comment, and we have now clearly outlined the arbitrary qualitative notation of x in Table1 legend.

Reviewer 2 Report
Comments and Suggestions for Authors
The manuscript was not improved enough.
1. Figure 1 still looks not professional.
2. Table 1 - it is hard to understand the Xs.
Author Response

(The authors gave the same response as above.)

Round 3
Reviewer 2 Report
Comments and Suggestions for Authors
The authors have not fully addressed my concerns.
First, the table misses the explanation what is referred as lower and higher toxicities.
Second, figure 1 from the scientific point looks like it is a drawing on the board but not a scientific figure. What is a huge red-lined tear-shaped object?
Otherwise the paper was improved.
Round 4
Reviewer 2 Report
Comments and Suggestions for Authors
The Authors addressed all my issues and I belive this paper may be accepted for publication in IJMS.